

# Long-term outcomes and predictors of patients with ST elevated versus non-ST elevated myocardial infarctions in non-obstructive coronary arteries: a retrospective study in Northern China

Lin Chen[1,2], Yinghong Fan[1,2], Zhen Fang[3] and Ning Liu[3]

[1] Institute of Gastroenterology, Affiliated Hospital of Yangzhou University, Yangzhou University, Yangzhou, Jiangsu Province, China

[2] Pancreatic Center, Department of Gastroenterology, The Affiliated Hospital of Yangzhou University, Yangzhou University, Yangzhou, Jiangsu Province, China

[3] Department of Cardiology, Northern Jiangsu People's Hospital, Yangzhou, Jiangsu Province, China

Corresponding author
Ning Liu, 844465791@qq.com

## ABSTRACT

**Background**. Myocardial infarction with non-obstructive coronary arteries (MINOCA) is a heterogeneous disease entity with diverse etiologies and no uniform treatment protocols. Patients with MINOCA can be clinically classified into two groups based on whether they have an ST-segment elevation (STE) or non-ST segment elevation (NSTE), based on electrocardiogram (ECG) results, whose clinical prognosis is unclear. This study aimed to compare the outcomes and predictors of patients with STE and NSTE in the MINOCA population.

**Methods**. We collected the data for 196 patients with MINOCA (115 with STE and 81 with NSTE) in China. Clinical characteristics, prognoses, and predictors of major adverse cardiovascular events (MACE) were analyzed during the follow-up of all patients.

**Results**. The proportion of patients with STE was greater than that with NSTE in the MINOCA population. Patients with NSTE were older and had a higher incidence of hypertension. No differences were observed in the outcomes between the STE and NSTE groups during a median follow-up period of 49 (37,46) months. No significant differences were observed in those with MACE (24.35% vs 22.22%, $P = 0.73$) and those without MACE. The multivariable predictors of MACE in the NSTE groups were Killip grades $\geq 2$ (HR 9.035, CI 95% [1.657–49.263], $P = 0.011$), reduced use of β-blockers during hospitalization (HR 0.238, CI 95% [0.072–0.788], $P = 0.019$), and higher levels of low-density lipoprotein cholesterol (LDL-C) (HR 2.267, CI 95% [1.008–5.097], $P = 0.048$); the reduced use of β-blockers during hospitalization was the only independent risk factor of MACE in the STE group.

**Conclusions**. There were differences between the clinical characteristics of patients with STE and NSTE in the MINOCA population, even though outcomes during follow-up were similar. Independent risk factors for major adverse cardiac events were not identical in the STE and NSTE groups, which could be attributable to the differences in disease pathogenesis.

## INTRODUCTION

Myocardial infarction with nonobstructive coronary arteries (MINOCA) is a heterogeneous group of diseases with different pathogeneses. It is characterized by acute myocardial infarction with normal coronary arteries or mild coronary artery stenosis (stenosis <50%), and occurs commonly in young women (*Tamis-Holland et al., 2019*). The prevalence of MINOCA reportedly ranges between 1–15% in patients with acute myocardial infarction (AMI), according to different studies (*Abdu et al., 2019*), and its overall prevalence was 6% in a recent meta-analysis (*Pasupathy et al., 2015*). MINOCA is a group of syndromes with multiple causes. Individuals with MINOCA can be classified into multiple subgroups, such as those with plaque rupture, coronary dissection, coronary artery spasm, and clinically unrecognized myocarditis or Takotsubo cardiomyopathy; all of these have different underlying pathophysiological mechanisms (*Agewall et al., 2017*; *Niccoli, Scalone & Crea, 2015*). Therefore, it is potentially challenging to effectively treat MINOCA patients for whom multiple pathogenic mechanisms have various underlying causes. The pathogenesis and prognosis of MINOCA patients need to be assessed further in future studies.

Previous studies have reported that patients with MINOCA had lower rates of major adverse cardiovascular events (MACE) and mortality during follow-up than patients with MI-CAD (*Montenegro Sá et al., 2018*; *Pasupathy et al., 2015*; *Pizzi et al., 2016*). Although patients with MINOCA appear to have a slightly better long-term prognosis, compared to MI-CAD (MI with obstructive coronary artery disease) patients, studies conducted in recent years have shown that MINOCA is not always benign (*Barr et al., 2018*; *Raparelli et al., 2018*). Notably, a Swedish study conducted over 4 years has shown that adverse cardiovascular events occurred in 23.9% of MINOCA patients during follow-up; among these, the mortality rate could be as high as 13.4% (*Lindahl et al., 2017*). Moreover, a Japanese study also showed that MINOCA patients had a higher mortality rate within 30 days of follow-up, as compared to MI-CAD patients (4.48% VS 3.46%) (*Ishii et al., 2020*).

However, the differences in clinical features and prognosis between patients with ST-segment elevated myocardial infarction (STEMI) and non-ST segment elevated myocardial infarction (NSTEMI) remain controversial. The occurrence of NSTEMI is more common than STEMI in the MINOCA population (*Pasupathy et al., 2015*; *Xu et al., 2020*). Previous studies have reported that STEMI patients had a poorer short-term prognosis and a more favorable long-term prognosis (*Borrayo-Sánchez et al., 2018*; *Polonski et al., 2011*). A large-scale Swedish study of MINOCA patients reported that during the 2.6-year follow-up period, the mortality rate for STEMI patients was 8%, while the mortality rate for NSTEMI patients was lower at 5% (*Nordenskjöld et al., 2018*), which was inconsistent with the results reported by *Li et al. (2022)*. Nevertheless, some studies suggest that there were no differences in prognosis between patients with STEMI and NSTEMI (*Montalescot et al., 2007*).

Some studies have shown that the history of atrial fibrillation, Killip grade, age, and treatment strategy were significant independent risk factors for prognosis in MINOCA patients (*Montalescot et al., 2007*; *Polonski et al., 2011*), while the predictors of prognosis in STE and NSTE patients are still unclear. Although the differences in prognosis between STEMI and NSTEMI patients in the AMI population have been reported hitherto, the differences in prognosis and predictors of prognosis among MINOCA patients with STE and NSTE remain unclear. This study aimed to compare the clinical features, prognosis, and predictors of MACE during the follow-up period among MINOCA patients with STE and NSTE in Northern China.

## MATERIALS & METHODS

### Patients

We conducted a retrospective study of patients who had been admitted to the First Hospital at Jilin University due to AMI from January 2015 to July 2018 and had undergone coronary angiography during hospitalization. Patients were included in the study if: (1) they met the diagnostic criteria specified in the AMI guidelines (*Thygesen et al., 2018*); (2) no occlusion of any infarct-related coronary artery and <50% stenosis could be observed in all epicardial vessels; (3) the patient received no other alternative diagnosis during clinical presentation (*e.g.*, non-ischemic causes such as sepsis, acute renal failure, pulmonary embolism, and myocarditis); and (4) age >18 years. Patients were excluded if: (1) thrombolytic therapy had been performed prior to coronary angiography; (2) they had a previous myocardial infarction or coronary revascularization; (3) previously underwent cardiac surgery; (4) had malignant tumors.

This study has been conducted in accordance with the Declaration of Helsinki and was approved by the Ethical Review Board of the hospital (the First Hospital of Jilin University, Changchun, China). Patient informed consent was waived, as this study was retrospective.

### Data collection

Most of the data were obtained from the medical records at the First Hospital of Jilin University that contained data on the baseline characteristics, biochemical markers, electrocardiogram (ECG) images, coronary angiography, and medications provided during hospitalization. Basic patient information (*e.g.*, age, sex) and past medical history (*e.g.*, smoking history, history of hypertension, hyperlipidemia, diabetes, arrhythmias) were recorded in detail. The history of arrhythmias including previous atrial arrhythmias or ventricular arrhythmias or heart block. We collected information regarding biochemical markers, including blood cardiac troponin-T(cTnT), creatine kinase-MB(CK-MB), brain natriuretic peptide (BNP), total cholesterol (TC), low-density lipoprotein cholesterol (LDL-C), high-density lipoprotein cholesterol (HDL-C), triglyceride (TG) and indicators of echocardiography, including LV (left ventricle) and LVEF (left ventricular ejection fraction) in 24 h after hospitalization. We classified the patients into the STE and NSTE groups based on their ECG results. STE and NSTE were defined in accordance with the Fourth Universal Definition of Myocardial Infarction (*Thygesen et al., 2018*).

The primary clinical endpoint of our study was the occurrence of major adverse cardiovascular events (MACE), including rehospitalization for increased chest pain that did not meet the criteria of AMI, based on ECG results and myocardial injury marker levels, and occurrence of non-fatal MI, heart failure, stroke, heart valve replacement, and all-cause deaths, which included cardiovascular and non-cardiovascular deaths. A diagnosis of MI was made if patients exhibited the dynamic development of cardiac troponin in conjunction with symptoms suggestive of myocardial ischemia. Cardiovascular death was defined as death because of acute coronary syndrome (ACS), cardiac rupture, severe arrhythmias, or refractory severe heart failure. A stroke was defined as an ischemic cerebral infarction caused by thrombotic or embolic occlusions in any major intracranial artery. A diagnosis of heart failure (HF) was established according to the current guidelines of the European Society of Cardiology (ESC) (*Ponikowski et al., 2016*).

## Statistical analysis

Statistical analysis was performed using SPSS 25.0 software. Normally distributed continuous variables were presented as mean ± standard deviation (SD) values. Non-normally distributed continuous variables were presented in terms of the median and inter-quartile range (IQR). Categorical variables were presented as counts and percentages. An independent sample *t*-test and the Mann–Whitney U test were used to perform a comparison of continuous variables between groups. Categorical variables were compared using the Chi-square and Fisher's exact tests. We used logistic regression analysis to evaluate the independent risk factors of outcomes in the STE and NSTE groups, while the adjusted OR for MACE was calculated *via* logistic regression analysis. All the tests performed were two-sided tests and values were identified to be statistically significant at a *P*-value <0.05.

## RESULTS

### Baseline characteristics of patients

In our study, the median follow-up period was 49 (37,46) months. A total of 9696 patients were diagnosed with MI; among these, 196 patients (2.02%) satisfied the diagnostic criteria for MINOCA. Based on the ECG results, 115 patients (58.7%) were included in the STE group, while 81 patients (41.3%) were included in the NSTE group (Fig. 1). A comparison of the baseline characteristics between patients with STE and NST among the MINOCA population has been shown in Table 1.

In comparison to NSTE patients, patients with STE were younger. Patients with NSTE had a higher incidence of hypertension, whereas no significant differences were observed in the incidence of other coronary risk factors (*e.g.*, diabetes, hyperlipidemia, previous arrhythmia, smoking history). The medications administered at discharge have been shown in Table 1. There were no significant differences between the two groups except for the fact that the more frequent use of aspirin and lower use of ACEI/ARB at admission in the STE group. Thus, the proportions of patients using clopidogrel, β-blockers, and statins were similar in the two groups. The level of serum glucose on admission in the NSTE group was higher than that in the STE group, while the other laboratory parameters were not significantly different among the two groups.

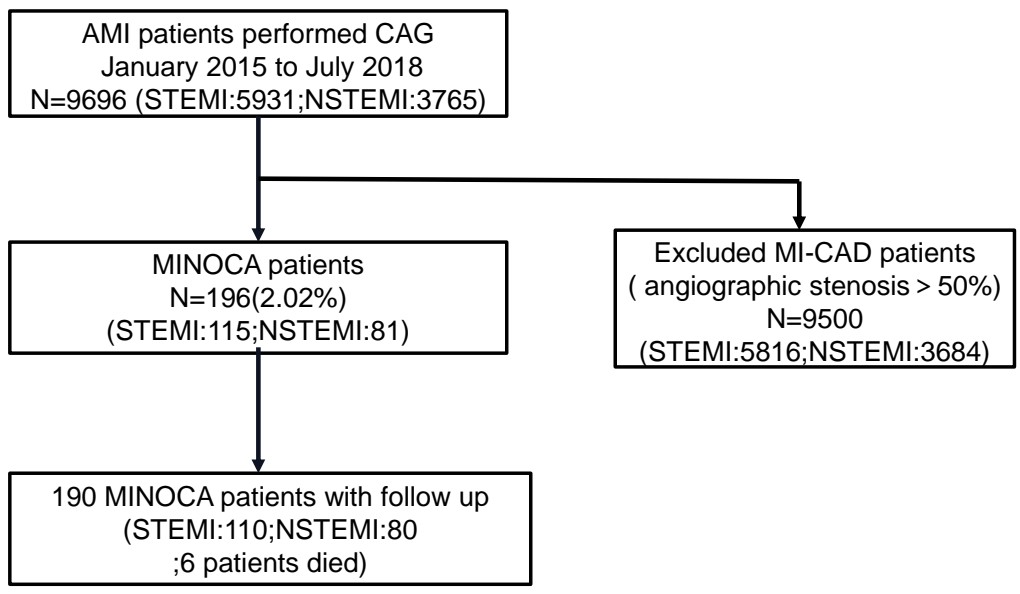

**Figure 1** **Flow chart of patients included in this study.** Flow chart of cases collection in this study.

## Follow-up

During a median follow-up period of 49 months (interquartile range [IQR] 37-61), MACE occurred in 46 (23.47%) out of a total of 51 patients. In the STE and NSTE groups, we observed the occurrence of MACE in 28 (24.35%) and 18 (22.22%) patients, respectively. The data are shown in Table 2. Thirty-one cases of MACE were observed in 28 patients (24.35%) in the STEMI group, including in patients who had to undergo rehospitalization for chest pain (four cases, 3.48%), non-fatal MI (three cases, 2.61%), heart failure (14 cases, 12.17%), stroke (five cases, 4.35%), and all-cause deaths (five cases, 4.35%). In the NSTEMI group, 20 cases MACE (24.35%) were observed in 18 patients (22.22%); these included chest pain (four patients, 4.49%, non-fatal MI (two patients, 2.47%), heart failure (eight patients, 9.88%), stroke (three patients, 3.70%), heart valve replacement (two patients, 2.47%), and all-cause death (one patient, 1.23%).

There were no statistical differences in the prevalence of MACE between the NSTE and STE groups ($P = 0.73$). In this study, five patients died of cardiogenic diseases. During the follow-up period, there was no significant difference in the incidence of chest pain, non-fatal MI, heart failure, stroke, heart valve replacement, and all-cause death between the STE and NSTE groups among the MINOCA population ($P > 0.05$).

## Predictive factors

Univariate analysis showed that older age, Killip grade $\geq 2$, longer hospitalization duration, being born male, lower use of β-blockers during hospitalization, and red blood cell counts were significant risk factors for MACE in the STE group (Table 3).

We conducted a multivariate analysis adjusted for age, Killip grades, hospitalization duration, sex, use of β-blockers during hospitalization, red blood cell counts, history of

**Table 1  Comparision of the baseline characteristics between STEMI and NSTEM among MINOCA population.**

| Variables | STEMI (n = 115) | NSTEMI (n = 81) | P |
|---|---|---|---|
| Demographics | | | |
| Age (years) | 52.93 ± 12.68 | 56.47 ± 11.21 | 0.045 |
| Male, n (%) | 81 (70.43) | 60 (74.07) | 0.577 |
| Coronary risk factors | | | |
| Diabetes, n (%) | 11 (9.57) | 12 (14.81) | 0.261 |
| Hypertension, n (%) | 47 (40.87) | 48 (59.26) | 0.011 |
| hyperlipidaemia, n (%) | 33 (28.70) | 23 (28.39) | 0.927 |
| previous arrhythmia, n (%) | 9 (7.83) | 7 (8.64) | 0.837 |
| Smoking history, n (%) | 83 (72.13) | 52 (64.20) | 0.235 |
| Killip grade, n (%) | | | |
| 1 grade | 103 (89.57) | 64 (79.01) | 0.040 |
| ≥2 grades | 12 (10.43) | 17 (20.99) | |
| hospitalization days (days) | 6 (4,8) | 7 (4,8) | 0.305 |
| Medications during hospitalization | | | |
| Aspirin, n (%) | 112 (97.39) | 73 (90.12) | 0.018 |
| Clopidogrel, n (%) | 108 (93.91) | 73 (90.12) | 0.415 |
| β-blocker, n (%) | 51 (44.35) | 41 (50.62) | 0.468 |
| Statins, n (%) | 110 (95.65) | 77 (95.06) | 0.721 |
| ACEI/ARB (%) | 44 (38.26) | 45 (55.56) | 0.017 |
| Laboratory indicators | | | |
| Myoglobin (ng/ml) | 94 (46.8,309.00) | 101.5 (53.08,178.75) | 0.917 |
| cTnT ((ng/ml)) | 3.02 (0.18,13.5) | 1.34 (0.22,5.87) | 0.076 |
| CK-MB | 5.95 (1.08,33,45) | 3.80 (1.00,12.81) | 0.102 |
| BNP | 112 (27.13,297.75) | 61.90 (20.40,186.00) | 0.106 |
| WBC count ($\times 10^{12}$/L) | 8.24 (6.24,10.38) | 7.52 (6.01,9.28) | 0.158 |
| NE (%) | 5.24 (3.94, 8.04) | 4.99 (3.71,6.55) | 0,231 |
| RBC count ($\times 10^{12}$/L) | 4.61 (4.27,4.97) | 4.69 (4.45,4.94) | 0.457 |
| PLT count ($\times 10^{12}$/L) | 221.5 (182.00,266.75) | 210 (172.25,255.75) | 0.262 |
| TC (mmol/L) | 4.20 (3.67,4.95) | 4.51 (3.75, 5.25) | 0.115 |
| LDL-C (mmol/L | 2.44 (1.99,1.45) | 2.45 (2.09,3.35) | 0.109 |
| HDL-C (mmol/L) | 1.18 (0.99,1.45) | 1.2 (1.05,1.48) | 0.305 |
| TG (mmol/L | 1.45 (1.06,2.37) | 1.51 (1.11,2.21) | 0.718 |
| Serum glucose (mmol/L) | 5.19 (4.66,5.95) | 5.55 (4.87,6.44) | 0.047 |
| Echocardiography | | | |
| LV (mm) | 49 (46, 52) | 50 (46.75,52) | 0.333 |
| LVEF (%) | 57 (54,60) | 59 (55, 60) | 0.128 |

**Notes.**
Abbreviation: cTnT, blood cardiac troponin-T; CK-MB, creatine kinase-MB; BNP, brain natriuretic peptide; RBC, Red blood cell; WBC, White blood cell; NE, neutrophilicgranulocyte; PLT, Platelet; TC, total cholesterol; LDL-C, low-density lipoprotein cholesterol; HDL-C, high-density lipoprotein cholesterol; TG, triglyceride; LV, left ventricle; LVEF, left ventricular ejection fraction.

**Table 2  Comparision of the rate of MACE in MINOCA during follow-up period.**

|  | STEMI (n = 115) | NSTEMI (n = 81) | P |
|---|---|---|---|
| MACE, n (%) | 28 (24.35) | 18 (22.22) | 0.73 |
| Chest pain rehospitalization, n (%) | 4 (3.48) | 4 (4.94) | 0.72 |
| nonfatal MI, n (%) | 3 (2.61) | 2 (2.47) | 1 |
| Heart failure, n (%) | 14 (12.17) | 8 (9.88) | 0.654 |
| Stroke, n (%) | 5 (4.35) | 3 (3.70) | 1 |
| Heart valve replacement, n (%) | – | 2 (2.47) | 0.17 |
| All-cause deaths, n (%) | 5 (4.35) | 1 (1.23) | 0.404 |

diabetes, and level of low-density lipoprotein cholesterol (LDL-C). The results showed that Killip grades $\geq 2$ (HR 9.035, Cl 95% [1.657–49.263], $P = 0.011$), lowered use of β-blockers during hospitalization (HR 0.238, Cl 95% [0.072−0.788], $P = 0.019$) and higher LDL-C levels (HR 2.267, Cl 95% [1.008−5.097], $P = 0.048$) were independent risk factors for MACE in patients with STE (Table 4).

Univariate analysis showed that older age and lowered use of β-blockers during hospitalization were associated with a higher extent of occurrence of MACE in the NSTE group (Table 3).

The age and extent of use of β-blockers and aspirin during hospitalization were adjusted *via* multivariate analysis. The results revealed that the lowered use of β-blockers during hospitalization was the only independent risk factor for MACE in patients with NSTE (HR 0.303, Cl 95% [0.093−0.991], $P = 0.048$). Thus, the use of β-blockers could improve the prognosis of MINOCA patients with NSTE.

## DISCUSSION

The objective of this study was to compare the prognosis and predictors of MACE among MINOCA patients with STE and NSTE. Our major findings were as follows: (1) There were differences in clinical features between the STE and NSTE groups among MINOCA patients; (2) there was no statistical difference in the incidence of MACE between the STE and NSTE groups during follow-up; (3) the independent risk predictors of MACE in MINOCA patients with STE include a higher level of LDL-C, Killip grades $\geq$2, and lowered use of β-blockers during hospitalization, whereas the lowered use of β-blockers during hospitalization was the only multivariable predictor of MACE in MINOCA patients with NSTE.

MINOCA has always been a confusing clinical entity that is characterized by myocardial infarctions with normal or near-normal coronary arteries of angiography (*Scalone, Niccoli & Crea, 2019*). Due to the difference in sample size and definition among various cohorts, the incidence of MINOCA in patients with acute myocardial infarction (AMI) is 1–15% (*Kilic et al., 2020*; *Nordenskjöld et al., 2018*; *Xu et al., 2020*), which is consistent with the findings of our study. Although the underlying causes of MINOCA are diverse, patients can be classified into the STEMI and NSTEMI groups based on their electrocardiogram (ECG) results. Among MINOCA patients, the proportion of patients with NSTEMI is

**Table 3  Univariate analysis of MACE among STEMI and NSTEMI population.**

| Factors | MACE$_{STE}$ ($n = 28$) | P$_{STE}$ | MACE$_{NSTE}$ ($n = 18$) | P$_{NSTE}$ |
|---|---|---|---|---|
| Age (years) | 57.39 ± 13.74 | 0.003 | 60.44 ± 9.94 | 0.049 |
| Male, n (%) | 15 (53.57) | 0.025 | 15 (83.33) | 0.376 |
| Diabetes, n (%) | 5 (17.86) | 0.086 | 5 (27.78) | 0.126 |
| Hypertension, n (%) | 14 (50.00) | 0.259 | 13 (72.22) | 0.204 |
| hyperlipidaemia, n (%) | 6 (21.43) | 0.266 | 3 (16.67) | 0.245 |
| previous arrhythmia, n (%) | 2 (7.14) | 1.000 | 3 (16.67) | 0.339 |
| Smoking history, n (%) | 19 (67.85) | 0.558 | 11 (61.11) | 0.757 |
| Killip grade, n (%) | | | | 0.145 |
|     1 grade | 20 (71.43) | 0.001 | 12 (66.67) | |
|     ≥2 grades | 8 (28.57) | | 6 (33.33) | |
| Laboratory indicators | | | | |
|     Myoglobin (ng/ml) | 97.25 (50.20,248.25) | 0.539 | 105.00 (53.63,173.25) | 0.934 |
|     cTnT (ng/ml) | 2.65 (0.07,14.57) | 0.661 | 0.54 (0.19,8.99) | 0.578 |
|     CK-MB (ng/ml) | 4.17 (1.00,25.55) | 0.481 | 3.79 (1.00,15.12) | 0.986 |
|     BNP | 120.00 (32.60,848.00) | 0.305 | 85.45 (31.55,249.45) | 0.150 |
|     WBC count ($\times 10^{12}$/L) | 8.20 (5.97,11.65) | 0.931 | 7.52 (5.59,8.79) | 0.461 |
|     RBC count ($\times 10^{12}$/L) | 4.43 (4.15,4.66) | 0.016 | 4.68 (4.43,4.93) | 0.773 |
|     PLT count ($\times 10^{12}$/L) | 219.00 (186.00,255.00) | 0.401 | 198.50 (154.75,250.50) | 0.450 |
|     TC (mmol/L) | 4.11 (3.35,4.94) | 0.276 | 4.68 (3.52,5.37) | 0.775 |
|     LDL-C (mmol/L) | 2.32 (1.77,2.80) | 0.085 | 2.63 (1.98,3.20) | 0.579 |
|     HDL-C (mmol/L) | 1.21 (1.02,1.60) | 0.372 | 1.31 (1.17,1.59) | 0.113 |
|     TG (mmol/L | 1.33 (0.90,1.73) | 0.170 | 1.36 (0.95,2.00) | 0.229 |
|     Serum glucose (mmol/L) | 5.46 (4.73,6.72) | 0.262 | 5.62 (4.84,6.74) | 0.775 |
| Echocardiography | | | | |
|     LV (mm) | 49.00 (46.00,52.00) | 0.989 | 50.50 (47.75,55.00) | 0.191 |
|     LVEF (%) | 58.00 (51.00,60.00) | 0.949 | 58.00 (50.75,60.50) | 0.330 |
| hospitalization days (days) | 7.00 (5.00,8.00) | 0.037 | 6.50 (3.75,9.00) | 0.991 |
| Medications during hospitalization | | | | |
|     Aspirin, n (%) | 28 (100.00) | 1.000 | 14 (77.78) | 0.086 |
|     Clopidogrel, n (%) | 26 (92.86) | 0.634 | 17 (94.45) | 0.677 |
|     β-blocker, n (%) | 8 (28.57) | 0.048 | 5 (27.78) | 0.028 |
|     Statins, n (%) | 28 (100.00) | 0.571 | 17 (94.45) | 1.000 |
|     ACEI/ARB (%) | 25 (89.28) | 0.754 | 16 (88.89) | 0.534 |

**Notes.**

Abbreviation: cTnT, blood cardiac troponin-T; CK-MB, creatine kinase-MB; BNP, brain natriuretic peptide; RBC, Red blood cell; WBC, White blood cell; PLT, Platelet; TC, total cholesterol; LDL-C, low-density lipoprotein cholesterol; HDL-C, high-density lipoprotein cholesterol; TG, triglyceride; LV, left ventricle; LVEF, left ventricular ejection fraction.

higher than that of those with STEMI (*Pasupathy et al., 2015*; *Xu et al., 2020*), which was in contrast to the findings of our study. This result may be attributable to the fact that our study is a single-center study with a small sample size. Certain previous studies have reported that there were significant differences in the clinical features of MINOCA patients with STEMI and NSTEMI (*Borrayo-Sánchez et al., 2018*; *Hanssen et al., 2012*). Recently, a

**Table 4  Multivariable predictors of MACE in STEMI patients.**

| Factors | OR | 95% Cl | P |
|---|---|---|---|
| Killip grade | 9.035 | (1.657,49.263) | 0.011 |
| $\beta$-blocker | 0.238 | (0.072,0.788) | 0.019 |
| LDL-C | 2.267 | (1.008,5.097) | 0.048 |

**Notes.**
Abbreviation: LDL-C, low-density lipoprotein cholesterol.

Chinese study on MINOCA reported that patients with NSTE were older, mostly female, and had a higher incidence of atrial fibrillation. Furthermore, patients with STE were more likely to have a history of smoking and a higher diastolic blood pressure, whereas there were no significant differences in the incidence of other risk factors for coronary problems (*e.g.*, hypertension, diabetes) between the two groups (*Xu et al., 2020*). Our study found that patients with NSTEMI had a higher age and a higher proportion of the patients had hypertensive disease, compared to STEMI patients, which was consistent with the findings of the study by *Johnston et al. (2015)*. Therefore, these differences may be associated with the different pathogeneses of the two groups; this needs to be confirmed in multi-center and prospective studies with a large sample size.

The prognostic differences between STEMI and NSTEMI patients in the MINOCA population remain controversial. Previous studies have reported higher short-term mortality in STEMI patients and higher long-term mortality in NSTEMI patients (*Borrayo-Sánchez et al., 2018*; *Johnston et al., 2015*; *Park et al., 2013*), which was also observed in the MINOCA population (*Nordenskjöld et al., 2018*). *Johnston et al. (2015)* reported that all-cause mortality was significantly higher in MINOCA patients with STEMI than in NSTEMI patients and that their long-term prognosis was poorer (*Johnston et al., 2015*). A recent study demonstrated that the mortality of patients with MINOCA presenting with STEMI was relatively high at 4.5% at year 1 (*Gue et al., 2019*). This might be related to the occurrence of congestive heart failure because of highly extensive and severe myocardial damage. However, no statistically significant differences in mortality were observed between STEMI and NSTEMI patients in this study. Our findings were similar to those of *Xu et al. (2020)* because we found that there was no statistical difference in the incidence of MACE (rehospitalization for chest pain, non-fatal MI, heart failure, stroke, heart valve replacement and all-cause death, *etc.*) in the follow-up period between the STEMI and NSTEMI groups, which may be related to the similar drug therapies administered to patients from different sub-groups.

There are several inconsistencies regarding the predictors of MACE in STEMI and NSTEMI patients in previous studies. One study reported that STEMI and NSTEMI patients differed significantly with regard to predictors of early and late-term mortality (*Park et al., 2013*; *Polonski et al., 2011*). In addition, the study conducted by Xu demonstrated that the predictors of MACE in MINOCA patients with STE and NSTE were different; the independent predictors of MACE in the NSTEMI group were age, lower level of TC, hypertension, and smoking history, and the strongest predictors in the STEMI group were reduced LVEF levels and a history of diabetes mellitus (*Xu et al., 2020*). A large

meta-analysis showed that a further reduction in LDL-C levels was effective in reducing the incidence of prognostic cardiovascular disease and stroke (*Baigent et al., 2010*), which was consistent with our findings, which showed that a higher LDL-C level was an independent risk factor for MACE in the STEMI group. The use of statins in patients with MINOCA for reducing the LDL-C levels and stabilizing and controlling coronary plaque progression has had a beneficial prognostic impact (*Choo et al., 2019*). *Johnston et al. (2015)* found that among STEMI patients, the all-cause mortality was significantly higher in females than in males, while this difference in mortality between the sexes was not observed in our research. We suggest that this complexity is reflective of the heterogeneous features of MINOCA in terms of STE and NSTE.

Currently, there is no uniform treatment for the MINOCA population. We found that β-blocker medication was a protective factor for MACE during the follow-up period in the MINOCA population with NSTEMI and STEMI, which is consistent with the findings of *Ciliberti et al. (2021)*. However, the findings of the study by Adbu showed that the treatment of MINOCA with statins and ACEI/ARB had long-term beneficial effects on the outcome, whereas β-blocker and DAPT treatment seemed to have no significant effect on the occurrence of MINOCA (*Abdu et al., 2019*). The administration of characteristic therapies is necessary for patients in whom the occurrence of MINOCA is attributable to different underlying mechanisms. All the above-mentioned studies suggest that the use of secondary preventative medications for cardiovascular disease may significantly improve the prognosis of the MINOCA population and should be advocated, but this needs to be confirmed in multicenter studies with longer follow-up periods.

### Limitations

There are several limitations to our study. One of the major limitations is that our study was a single-center retrospective study with a small sample size and a short follow-up period, because of which our findings might lead to biased findings. Second, cardiac magnetic resonance (CMR) cloud not be performed for all patients due to medical insurance-related issues and the lack of CMR may influence the accuracy of our findings in MINOCA patients. Finally, information regarding medications to be administered in the follow-up period could not be obtained for all patients. Hence, we could not further analyze whether the long-term use of secondary preventative medications was beneficial for patients with MINOCA. A larger multi-center randomized controlled study is necessary to clarify the results of this study.

## CONCLUSIONS

In conclusion, the clinical characteristics of the STE and NSTE groups differed in patients with MINOCA, whereas the outcomes during the 49-month follow-up were similar. The predictors for MACE in patients between the STE group and NSTE group were not thoroughly identical.

### Funding
The authors received no funding for this work.

### Competing Interests
The authors declare there are no competing interests.

### Author Contributions

- Lin Chen conceived and designed the experiments, performed the experiments, analyzed the data, prepared figures and/or tables, authored or reviewed drafts of the article, and approved the final draft.
- Yinghong Fan performed the experiments, analyzed the data, prepared figures and/or tables, and approved the final draft.
- Zhen Fang performed the experiments, analyzed the data, authored or reviewed drafts of the article, and approved the final draft.
- Ning Liu conceived and designed the experiments, performed the experiments, analyzed the data, prepared figures and/or tables, authored or reviewed drafts of the article, and approved the final draft.

### Human Ethics

The following information was supplied relating to ethical approvals (i.e., approving body and any reference numbers):

the First Hospital of Jilin University

### Data Availability

The raw data is available in the Supplementary Files.

### Supplemental Information

Supplemental information for this article can be found online at http://dx.doi.org/10.7717/peerj.14958#supplemental-information.

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
