# Peer review of "Long-term outcomes and predictors of patients with ST elevated versus non-ST elevated myocardial infarctions in non-obstructive coronary arteries: a retrospective study in Northern China"

_PeerJ, doi:10.7717/peerj.14958_

## Round 0.1 · original submission · Major Revisions

Please address reviewers comments.

Reviewer 1 ·

Basic reporting

Clear language, easily understandable. Sufficient background and literature reference provided. Data shown clearly in the tables.

Suggestions -
Line 54 – MI with non "OBSTRUCTIVE" – Obstructive workd is missing
Line 66,68,72- MI- CAD ( what this abbreviation stand for)

Experimental design

In scope of the journal. Clearly defined research question, certainly fills the gap in the knowledge of MINOCA. Research method described in stepwise manner with ease of replicating study if desired.

Validity of the findings

Data provided appears robust with relevant statistical analysis. Clear conclusion explaining how this study adds to the existing literature and also raises the questions that additional research is needed to find better answer for the given condition.

Additional comments

Data suggested beta blocker at discharge but abstract says beta blocker "during hospitalization" ( line 44)
Please clarify.

Reviewer 2 ·

Basic reporting

I commend the authors for presenting this data of ~4-year follow-up of 196 patients with MINOCA with AMI either STE or NSTEMI with good inclusion and exclusion criteria. They excluded type 2 MI etiologies in this study like sepsis, acute renal failure, pulmonary embolism and myocarditis. 115 patients have STE, 59% of the population which is higher compared to previous studies; previous studies reported 1/3 of the MINOCA has STE on the ECG. Here are some of the observations noted or need more data:
1. Didn't mention how many have IVUS or OCT during the study to differentiate etiology of MI (Plaque rupture or erosion or dissection or thromboembolism or spasm or microvascular dysfunction)
2. Didn't include medications like ACEI/ARB/ARNI/CCB as significant % of patients in both groups have HTN. In a SWEDEHEART registry study of more than 9,000 MINOCA patients followed for 4 years, a propensity score model showed significantly reduced major adverse cardiovascular events (MACE) (composite of all-cause mortality, hospitalization for MI, ischemic stroke, and heart failure) in patients treated with statins or angiotensin-converting enzyme inhibitors/angiotensin receptor blockers.
3. Line 29: mentioned there are no uniform treatment protocols. The following article highlights the diagnostic algorithm for MINOCA helps to guide the etiology of the MINOCA and treatment varies based on the etiology. Agewall S Beltrame JF, Reynolds HR, et al. ESC working group position paper on myocardial infarction with non-obstructive coronary arteries. Eur Heart J 2017;38:143-53.
4. Line 251: typo MONICA instead of MINOCA
5. "Previous arrhythmias" does it include atrial arrhythmias or ventricular arrhythmias or heart block
6. How many of them have LV function less than 35% and how many have "defibrillators" or CRT-D; might be very little number
7. CMR is a crucial part of work up for MINOCA as it would have helped to differentiate further diagnosis like Myocarditis, Takostubo or normal myocardium. Authors kindly mentioned this in the discussion part due to lack of insurance and lack of CMR availability.
8. Grammatical errors: include "space" after comma (,) or full stop (.)

Overall, my impression is the paper is at a very "superficial" level but could improve if they can comment or gather more data on those comments I mentioned above. I would probably on the side of "Accept with major revision"

Experimental design

Please see above

Validity of the findings

Please see above

---

## Round 0.2 · Minor Revisions

Dear authors, as an Editor I have one question in addition to the ones already addressed.

How was MINOCA diagnosed in this group?

---

## Round 0.3 · Minor Revisions

This was a retrospective review of medical records. Ethical approval was not obtained until January 2021. Please clarify how the consent was obtained from the participants, or remove this statement and obtain a waiver of consent from the institutional IRB

---

## Round 0.4 · Major Revisions

It would be unacceptable to publish an unapproved statement that verbal consent was obtained for study participation. The Declaration of Helsinki states that non-written consent must be formally documented and witnessed (and should only occur if written consent is not possible).

Therefore, we require IRB approval for waiver of written consent, or a written consent from the participants.

---

## Round 0.5 · Minor Revisions

Thank you for the changes. Please omit 'telephone conversations and clinic visits' as this is a purely retrospective study.

---

## Round 0.6 · accepted · Accept

Congratulations. Your article has been accepted.